

# Interactive effects of drought and deforestation on multitrophic communities and aquatic ecosystem functions in the Neotropics—a test using tank bromeliads

Marie Séguigne[1], Céline Leroy[2,3], Jean-François Carrias[4], Bruno Corbara[4], Tristan Lafont Rapnouil[1,2,3] and Régis Céréghino[1]

[1] Centre de Recherche sur la Biodiversité et l'Environnement (CRBE), Université de Toulouse, CNRS, IRD, Toulouse INP, Université Toulouse 3—Paul Sabatier (UT3), Toulouse, France
[2] AMAP, Université de Montpellier, CIRAD, CNRS, INRAE, IRD, Montpellier, France
[3] EcoFoG, AgroParisTech, CIRAD, CNRS, INRAE, Université des Antilles, Université de Guyane, Campus agronomique, Kourou, France
[4] Laboratoire Microorganismes: Génome et Environnement (LMGE), Université Clermont Auvergne, CNRS, F-63000, Clermont-Ferrand, France

Corresponding author
Marie Séguigne,
marie.seguigne@univ-tlse3.fr

## ABSTRACT

**Background:** Together with the intensification of dry seasons in Neotropical regions, increasing deforestation is expected to exacerbate species extinctions, something that could lead to dramatic shifts in multitrophic communities and ecosystem functions. Recent studies suggest that the effects of habitat loss are greater where precipitation has decreased. Yet, experimental studies of the pure and interactive effects of drought and deforestation at ecosystem level remain scarce.

**Methods:** Here, we used rainshelters and transplantation from rainforest to open areas of natural microcosms (the aquatic ecosystem and microbial-faunal food web found within the rainwater-filled leaves of tank bromeliads) to emulate drought and deforestation in a full factorial experimental design. We analysed the pure and interactive effects of our treatments on functional community structure (including microorganisms, detritivore and predatory invertebrates), and on leaf litter decomposition in tank bromeliad ecosystems.

**Results:** Drought or deforestation alone had a moderate impact on biomass at the various trophic level, but did not eliminate species. However, their interaction synergistically reduced the biomass of all invertebrate functional groups and bacteria. Predators were the most impacted trophic group as they were totally eliminated, while detritivore biomass was reduced by about 95%. Fungal biomass was either unaffected or boosted by our treatments. Decomposition was essentially driven by microbial activity, and did not change across treatments involving deforestation and/or drought.

**Conclusions:** Our results suggest that highly resistant microorganisms such as fungi (plus a few detritivores) maintain key ecosystem functions in the face of drought and habitat change. We conclude that habitat destruction compounds the problems of climate change, that the impacts of the two phenomena on food webs are mutually reinforcing, and that the stability of ecosystem functions depends on the resistance of

a core group of organisms. Assuming that taking global action is more challenging than taking local-regional actions, policy-makers should be encouraged to implement environmental action plans that will halt habitat destruction, to dampen any detrimental interactive effect with the impacts of global climate change.

# INTRODUCTION

Decades of overharvesting, habitat destruction, biological invasions, ecosystem contamination and emission of greenhouse gases have led to the degradation of ecosystems worldwide (*Maxwell et al., 2016*; *Urban et al., 2016*). In particular, it is broadly acknowledged that habitat modification and climate change are among the greatest threats to biodiversity and ecosystem functions (*Rice et al., 2018*), and that the two phenomena are mutually reinforcing (*Brook, Sodhi & Bradshaw, 2008*; *Lorenzen et al., 2011*). The local to global components of environmental change (*e.g.*, deforestation, precipitation change) are often studied in isolation (*e.g.*, *Bennett et al., 2019*; *Foden et al., 2019*; *Magioli et al., 2019*; *Nowakowski et al., 2017*) though there is growing evidence of complex interactive effects between the various drivers of biodiversity decline (*Murdoch, Mantyka-Pringle & Sharma, 2020*; *Oliver & Morecroft, 2014*; *Staudt et al., 2013*). In most cases, the combined effects of multiple stressors on any given environmental variable are not simply additive and can result in an "ecological surprise" (*Fong, Bittick & Fong, 2018*; *Paine, Tegner & Johnson, 1998*). The interactive effects of stressors can be either greater (synergistic) or weaker (antagonistic) than the simple sum (additive) of the effects induced by individual stressors (*Côté, Darling & Brown, 2016*; *Orr et al., 2020*; *Piggott, Townsend & Matthaei, 2015*). To date, this question has been essentially examined in temperate regions of the northern hemisphere, leaving tropical regions overlooked (*Oliver & Morecroft, 2014*; *Tyukavina et al., 2017*). Paradoxically, tropical ecosystems could lose more species than their temperate counterparts, first because they contain a disproportionate number of the world's species (*Barlow et al., 2018*), and second, because tropical species often have narrower physiological tolerances for climatic variation (*Tewksbury, Huey & Deutsch, 2008*).

Freshwaters are the most endangered ecosystems of the World (*Jenkins, 2003*). Threats include human activities that have local to regional impacts, to which global-scale stressors are superimposed (*Dudgeon et al., 2006*). These issues are particularly concerning in Neotropical regions where, together with increasing rates of deforestation and land conversion (*Ríos-Touma & Ramírez, 2019*; *Schielein & Börner, 2018*), the intensification of dry seasons is expected to exacerbate extinctions by reducing water availability (*Seneviratne et al., 2021*). On one hand, deforestation opens the canopy thus modifying food availability and increasing incident light in freshwater ecosystems. Under these circumstances, we expect algae to grow and proportions of invertebrate shredders should fall due to lower inputs of coarse detritus biomass (*Barlow et al., 2018*; *Brito et al., 2020*;

*Silva-Araújo et al., 2020*). Thus, either grazer-scraper and collector taxa should dominate the invertebrate biomass (*Brouard et al., 2012*). On the other hand, different functional groups or biotic compartments could respond differently to drought-induced stress (*Srivastava et al., 2020*), leading to reconfigurations of the multitrophic communities. Owing to their larger body size and energetic demands, macro-organisms should notably show stronger responses, compared to the microorganisms (*Vilmi et al., 2020*). Finally, if the nature of the interaction between deforestation and drought (*i.e.*, additive *vs* synergistic) and the subsequent effects on ecosystem functions remain to be specified, the interactive effects should be much stronger than the pure effects of each disturbance type in isolation. Indeed, food web-mediated effects due to deforestation such as the loss of entire functional groups (*e.g.*, leaf shredders) will alter processing chains and species interactions, further weakening ecosystems in the face drought. Understanding the impacts of local and global disturbance, and perhaps more importantly their interactive effects on the structure of multitrophic aquatic communities is therefore an important step towards predicting the ecosystem-level consequences of environmental changes in the Neotropics.

Whilst experiments are widely used to test specific hypotheses regarding the effects of environmental change, emulating effects of local (*e.g.*, deforestation) and global stressors (*e.g.*, precipitation change) at the whole-community and/or -ecosystem level in natural environments is challenging. For instance, drought cannot be manipulated at the extent of large macrocosms, and natural ecosystems are usually too diverse in size or shape to form actual replicates. Natural microcosms such as plant-held waters and their aquatic communities are relevant model system to test ecological hypotheses in experimental research (*Srivastava et al., 2004*). Native to the Neotropics, the Bromeliaceae family comprises more than 3,700 species of flowering plant, half of them being tank-forming bromeliads (*Givnish et al., 2014*). Tank bromeliads have leaves arranged in rosettes that collect rainwater (up to several litres) and litter from overhanging trees. These tanks are a suitable habitat for specialized aquatic organisms ranging from prokaryotes to predatory invertebrates, assembled in multi-trophic communities (*Brouard et al., 2012*). Tank bromeliads are highly replicated in nature and span a broad range of environments. Because they can be easily transplanted (the roots primarily have and anchoring role), can be exhaustively sampled, and host co-evolved species, tank bromeliads were deemed relevant systems to test the effects of multiple stressors at ecosystem level in manipulative experiments conducted in nature (*Céréghino et al., 2022*; *Romero et al., 2020*; *Srivastava et al., 2020*; *Trzcinski et al., 2016*).

In this study, we used the tank bromeliad ecosystem to tease out the pure and interactive effects of deforestation and drought on freshwater food webs and ecosystem functions in Neotropical environments. Specifically, we used cross-habitat transplantation and rainout shelters to examine how emulated canopy trimming and drought, and their interaction, affect the various functional components of aquatic communities, and subsequently, ecosystem functions. We used detrital decomposition, a key ecosystem process in nutrient and carbon cycling, as a proxy of ecosystem functioning (*Benfield, Fritz & Tiegs, 2017*; *Gessner et al., 2010*). We tested three hypotheses. First, if understorey environments
determine the main energy pathways in aquatic ecosystems (*Brouard et al., 2012*; *Neres-Lima et al., 2017*), then deforestation alone should primarily impact the "brown", benthic to pelagic energy pathway (*Lorion & Kennedy, 2009*). More specifically, we predicted a decline in the overall biomass of detritivores, including macro- and micro-organisms that breakdown coarse particulate organic matter (CPOM), and those that process the resulting fine particulate organic matter (FPOM). As a result, we'd expect decomposition to be adversely affected by deforestation (*Silva-Junior et al., 2014*). Second, if water shrinkage selects against organisms with a large body size and higher energetic demand (*Ledger et al., 2013*; *Ruiz et al., 2022*) and/or with a pelagic life style (*Aspin et al., 2019*), then drought alone should cause a decline in predator biomass while favouring benthic detritivores. Subsequently, predation release could foster detritivore activity and, therefore, drought could indirectly increase detrital decomposition (*Amundrud & Srivastava, 2016*). Third, if deforestation and drought disassemble food webs both from the bottom and from the top (see above predictions), then we expect their interactive effects to be much stronger than the sum of their pure effects because the disruption of trophic interactions and processing chains will prevent shifts to alternative states and cause a collapse of communities and ecosystem functions.

## MATERIALS AND METHODS

### Study area and bromeliads

Our experiment was conducted in French Guiana, near the Petit-Saut Dam (5°04′39″N, 52°59′11″W; elevation <80 m), from October 2021 to March 2022. The area is under a tropical wet climate with low seasonal variations of temperature (monthly average = 20.5–33.5 °C). Humidity ranges from 70 to 100%, with 3.000 mm of annual precipitation. A dry season occurs from September to November and there is a short dry period in March. The remaining months correspond to the rainy season. The average number of consecutive days without rainfall in the area is 26 ± 5.3 days (annual mean ± SD, based on daily rainfall records over the past 20 years at the Paracou weather station, 8 km away from our field site). We selected two habitats: a closed rainforest (here after "forest"), and a sun-exposed area located 50 m away from the edge of the forest (here after "open"). *Aechmea aquilega* (Salib.) Griseb. is the dominant tank bromeliad species in the area and was present in both the forest and open area. This bromeliad has large reservoirs that ease manipulation of flow-through enclosures (*Carrias et al., 2020*; *Trzcinski et al., 2016*).

### Leaf litter and flow-through enclosures

We chose a species of tree commonly found in forests of French Guiana, *Goupia glabra* (Goupiaceae), known for the rapid decomposition of its leaf litter in its natural environments (*Coq et al., 2010*; *Hättenschwiler et al., 2011*). Freshly fallen leaves were collected from a single tree using litter traps in our study area. All leaves were hydrated with rain water and then cut into strips of 1 × 5 and 1 × 4 cm avoiding the central vein. All strips were sterilized with 70° ethanol and ultraviolet light to eliminate any prior surface microbiota. Then, the strips were oven dried at 40 °C for 48 h. The smaller strips

(1 × 4 cm), used to monitor leaf litter mass loss, were weighted to the nearest 0.1 mg. Following *Rodríguez-Pérez et al. (2018)*, we constructed flow-through enclosures from injection anchor sleeves (Fisher®, diameter = 16 mm, length = 85 mm, mesh size = 2 mm). Half of the enclosures were covered with 80 μm Nitex® mesh to create "fine-mesh enclosures" that excluded invertebrates while allowing water flow. The other half was not covered with Nitex, and additional holes were drilled at their surface (diameter = 4 mm). These "coarse-mesh enclosures" allowed movements of invertebrates. The use of these two mesh sizes enabled us to separate the contribution of microorganisms (fine mesh) from the overall biologically-assisted decomposition (coarse mesh). Each enclosure received three leaf strips: one 1 × 4 cm for decomposition, and two 1 × 5 cm to quantify the attached bacterial and fungal biomass. In total, we prepared 140 enclosures (70 fine- and 70 coarse-mesh enclosures) prior to their incubation in the 70 bromeliads used in the experiment (see below).

## Experimental set up

We tested the pure and interactive effects of deforestation and drought, including sequential and simultaneous effects, on functional community structure and leaf litter decomposition. Prior to the experiment, we selected 50 and 20 mature *A. aquilega* that naturally grew in the forest and open area, respectively, to end up with ten replicates per treatment (see below). All plants had similar vegetative traits (number of leaves = 13 ± 3, plant diameter = 107.9 ± 38.21 cm, water holding capacity), to avoid habitat-size effects on community diversity (*Petermann et al., 2015*). These plants were never used in any experiment before. The experiment started with the pure effect of deforestation. First, twenty bromeliads of the forested area were transplanted to the open area to emulate a deforestation event. Two enclosures (one fine- and one coarse-mesh enclosure) were immersed in an intermediate well of each of the 70 plants. After 105 days, we subjected bromeliads to a drought event of 65 days, which corresponds to the most extreme event recorded in the area over the past 20 years. To emulate drought, we placed transparent rainshelters 1 m above the plants. These rainshelters prevent natural water inputs without affecting macroinvertebrate colonisation or increasing air temperature (*Marino et al., 2017*). Ten of the transplanted bromeliads (sequential effect), 10 bromeliads in the forest as well as ten bromeliads in the open area were exposed to the drought event starting at day 105. An additional set of ten bromeliads were transplanted from the forest to the open area and directly exposed to drought (simultaneous effect). Finally, 10 unmanipulated plants in the forest and in the open area were used as controls. The percentages of total incident radiation above the bromeliads (IR) were calculated using hemispherical photographs and an image processing software (Gap Light Analyzer 2.0) as described in *Leroy et al. (2009)*. In total, the experiment comprised seven treatments with ten replicates each (Fig. 1): forest control (T1), deforestation (T2), drought in the forest (T3), deforestation followed by drought (T4), deforestation and drought simultaneously (T5), open area control (T6), and drought in the open area (T7). Interactive effects were tested both sequentially (T4) and simultaneously (T5) to further examine whether a recovery time between two disturbances affects the extent ecosystem responses (*Gunderson, Armstrong & Stillman, 2016*).

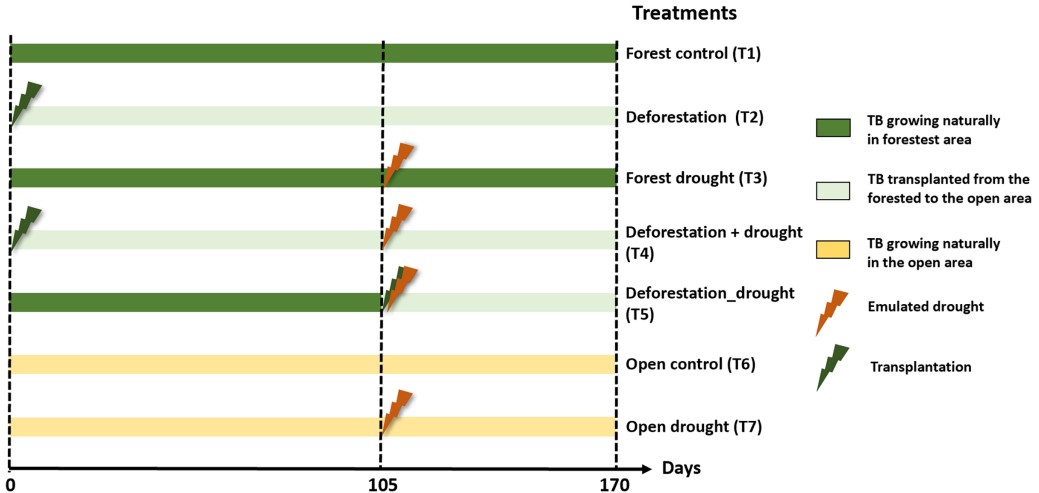

**Figure 1 Experimental design to test the interactive effects of drought and deforestation on aquatic communities and functions with seven treatments and 10 replicates per treatments.** Transplantation of bromeliads to the open area emulated deforestation, drought was emulated using rainout shelters above individual plants. TB, tank bromeliads.

## Sampling

At the end of the experiment (day 170), we collected the enclosures and the whole water content of each plant. One of the two $1 \times 5$ cm strips per enclosure was immediately preserved in 4% formaldehyde (final concentration). The other two leaf strips were conditioned in small plastic bags and kept in a cooler until processing in the laboratory. The plant content (water, organisms, and detritus) was then sucked out using micropipettes, the water volume collected (WV, mL) was measured with a graduated cylinder. The whole content was kept in pots and brought to the laboratory. All plant content collection were obtained under the French Ministry of Ecological Transition and Solidarity permit (ABSCH-IRCC-FR-247227-1).

## Laboratory analyses

The $1 \times 4$ cm leaf strips were oven dried at 60°C for 48 h and weighted to the nearest 0.1 mg. We used leaf mass loss data to estimate the decomposition rate $k$ (d$^{-1}$) based on the log-transformed exponential model (*Olson, 1963*): $k = -ln(m_t/m_0)/t$ where $m_0$ is the initial leaf mass, $m_t$ is the mass after decomposition and $t$ relates to the time in day (here 170 days). The $1 \times 5$ cm strips preserved in formaldehyde were sonicated ($3 \times 30$ s) using a sonication bath in order to improve cell detachment. The bacterial suspensions were then centrifugated at 800 g for 60 s and the supernatants diluted 10-fold and stained with the nucleic acid dye SYBR Green I (S7563; Thermo Fischer Scientific, Waltham, MA, USA). Counting of attached bacterial cells were then performed by flow cytometry (Plateforme CYSTEM–UCA PARTNER, Clermont-Ferrand, FRANCE) using a FACS Aria Fusion SORP flow cytometer (BD Biosciences, Franklin Lakes, NJ, USA) equipped with a 70 μm nozzle and a 1.5 neutral density filter to estimate a mean biovolume of bacteria. Considering that 10 μm$^3$ = 0.4 pg of bacteria dry weight (*Norland, Heldal & Tumyr, 1987*),

we calculated for each sample its associated attached bacterial biomass expressed in µg dry weight per gram dry weight of litter.

The fungal biomass was measured through ergosterol content, on the other 1 × 5 cm strips, by lipid extraction and High-Performance Liquid Chromatography (HPLC). We followed (*Gessner & Schmitt, 1996*) method and used a Agilent 1260 Infinity II system (Agilent Technologies, Santa Clara, CA, USA) for the analyses. Biomass of attached fungi was expressed as the concentration of ergosterol in µg per gram of litter dry weight.

Aquatic macroinvertebrates were sorted, identified to species or morphospecies, and enumerated. We used allometric relationships developed by us (*Dézerald et al., 2017*) to transform species counts into biomass. Invertebrate species biomass was aggregated by feeding guilds *sensu* *Merritt, Cummins & Berg (2017)* and *Merritt & Cummins (1978)*, including predators, prey, leaf shredders + leaf scrapers (feed on coarse particle organic matter, CPOM), and filter-feeders + gathering-collectors (feed on fine particle organic matter, FPOM, produced by the activity of shredders and scrapers).

## Data analysis

All analyses were performed with R software version 4.1.1 (*R Core Team, 2021*). Because assumptions of parametric tests were not generally met, even after usual data transformations (log- or square root-transformations), we used non-parametric approaches for our study response variable.

We first evaluated the main impact of deforestation and drought on habitat conditions by comparing variation in water volume (WV) and incident radiation (IR) among control and treatment bromeliads. Multiple pairwise comparisons with Bonferroni correction were implemented using the emmeans package after aligned rank transformation of data with the *ARTool* package (*Wobbrock et al., 2011*).

We then analysed the effect of the various treatments on the biomass of invertebrate feeding guilds. For each group, we conducted three sets of analyses: (i) comparison of controls (forest *vs* open) to evaluate differences under natural conditions, (ii) comparison of bromeliads originally present in the forest against forest control (T1 to T5) to evaluate the effects of deforestation, drought, and their interaction (where the interaction is directly represented by treatments T4 and T5 that combine both disturbances), and (iii) comparison of drought treatment and control in the open area (T6 *vs* T7). Comparisons (i) and (iii) where conducted using the Mann-Whitney U test adapted for small numbers of samples. Comparisons (ii) were conducted using Aligned rank transformed Anova with "treatment" as factor, which had five levels: "forest control", "deforestation", "forest drought", "deforestation-drought sequentially" and "deforestation-drought simultaneously". Then *post-hoc* contrast tests with the Dunnett method and Bonferroni correction were run to compare the effect of each treatment to the control group using *emmeans* package. Third, we analysed the effects of treatments on the biomass of microorganisms (bacteria and fungi) and litter decomposition. During the sorting process, some early-instar shredders were found in 11 fine-mesh enclosures from four treatments. To avoid any bias in our results, we removed those fine mesh enclosures from this specific analysis. The resulting dataset included six replicates for "forest control" and "deforestation", nine replicates for

"forest drought", and eight replicates for "deforestation-drought simultaneously" for fine mesh enclosures. We kept the 10 replicates in all other treatments. We initially ran comparisons (i), (ii) and (iii) using two-way ANOVA (aligned rank transformed) with the factors "treatments" and "mesh size", and their interactions. However, these models did not met assumptions of the Aligned Rank Transform ANOVA (aligned responses did not sum to zero). We therefore ran separate models for fine mesh and coarse mesh enclosures, using the same procedures described for invertebrate feeding guilds.

In all analyses, effects were considered as "significant" when p-values were inferior to 0.05, and "marginal" when p-values were inferior to 0.1.

## RESULTS

### Effects of deforestation and drought on habitat conditions

Control bromeliads from the forest held significantly less water, compared to control bromeliads from the open area (pairwise *post-hoc* tests; $p < 0.0001$) (Fig. 2, Table S1). In the forest, drought significantly reduced the water volume (WV) held by bromeliads (mean ± SE values: control forest = 512 ± 99 mL, drought = 48 ± 16 mL; $p < 0.0001$). The transplantation of forest bromeliads to the open area ("deforestation") did not significantly alter WV (deforestation group = 562 ± 64 mL, $p > 0.0500$). However, the combination of drought and deforestation dried out almost all transplanted bromeliads (WV = 0 mL in each bromeliad, except for one plant subjected to simultaneous disturbances), so there was highly significant difference in pairwise comparisons testing combined disturbances *vs* control or *vs* drought alone in the forest ($p < 0.0001$ for all). In the open area, WV was significantly reduced by drought (control open = 1,490 ± 242 mL, drought = 149 ± 45 mL; $p < 0.0001$). The mean incident radiation was 58 ± 4% in the open area, significantly higher than that recorded in the forest (19 ± 1%; $p < 0.0001$), (Fig. 2, Table S1).

### Invertebrate biomass

The list of invertebrates and associated feeding groups is given in Table S2. The comparison of control groups of bromeliads (forest *vs* open) showed that the biomass of predators did not differ significantly between habitats (Mann-Whitney U: W = 45.5, $p > 0.0500$), and that the biomass of detritivorous prey was significantly lower in the open area (Mann-Whitney U: W = 87, $p < 0.0100$; Fig. 3A).

Deforestation did not alter predator biomass (Table 1, $p > 0.0500$), but significantly decreased the biomass of detritivorous prey (Table 1, $p < 0.0500$). Drought did not significantly impact prey biomass in the forest (Table 1, $p > 0.0500$), and in the open area (Mann-Whitney U: W = 73, $p > 0.0500$). However, drought significantly decreased the biomass of predators, both in the forest (Table 1, $p < 0.0010$) and in the open area (Mann-Whitney U: W = 81.5, $p < 0.0500$). Finally, the interactive effect of deforestation and drought almost eliminated the predator species (Table 1, $p < 0.0001$), and dramatically decreased the biomass of prey (Table 1, $p < 0.0001$, Fig. 3A, see also Figs. S1 and S2). Overall, prey that were able to resist and maintain populations in bromeliads subject to
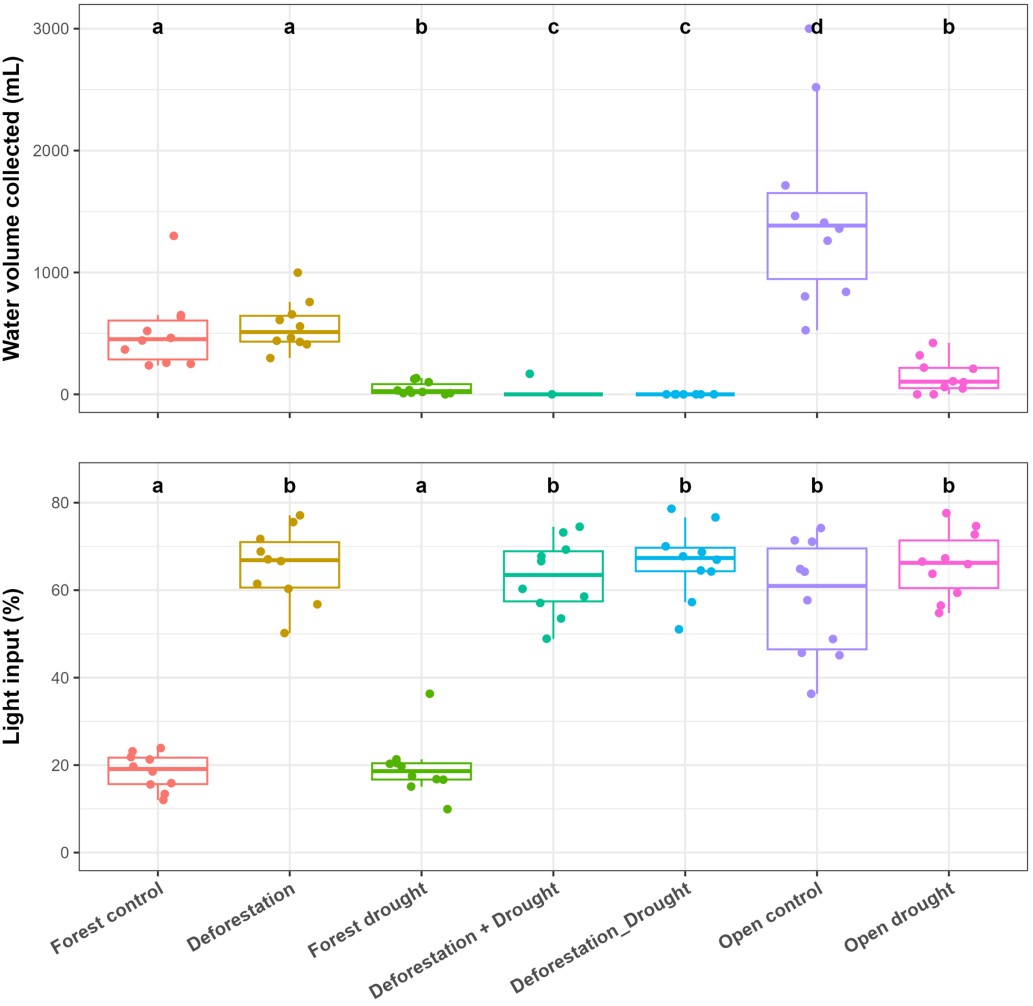

**Figure 2 Influence of the various treatments on the water volume held by bromeliads at the end of the experiment (WV, mL) and on incident radiation (IR, %) measured above the bromeliads.** Lowercase letters indicate significant differences ($p < 0.05$, Aligned rank transformed ANOVA and pairwise *post-hoc* tests, see Table S2).

deforestation x drought were *Trentepohlia* sp. (shredder), Chironomini, *Aulophorus superterrenus* and *Elpidium bromeliarum* (collector-gatherers) (Fig. S2).

When detritivores were further categorized by the size of ingested particles, the biomass of FPOM and CPOM consumers was significantly higher in the forested area compared to the open area (Fig. 3B; FPOM consumers, Mann-Whitney U: W = 77, $p < 0.0500$; CPOM consumers, Mann-Whitney U: W = 83.5, $p < 0.0500$). Neither deforestation nor drought alone significantly altered the biomass of FPOM and CPOM consumers (Table 1, $p > 0.0500$ in all tests). Similarly, FPOM and CPOM consumers were not significantly impacted by drought in the open area (FPOM consumers, Mann Whitney U: W = 67, $p > 0.0500$; CPOM consumers, Mann-Whitney U: W = 53.5, $p > 0.0500$). However, the combined effect of deforestation and drought, either sequentially or simultaneously, had a strong negative impact on the biomass of both groups (Table 1, $p \leq 0.0001$).

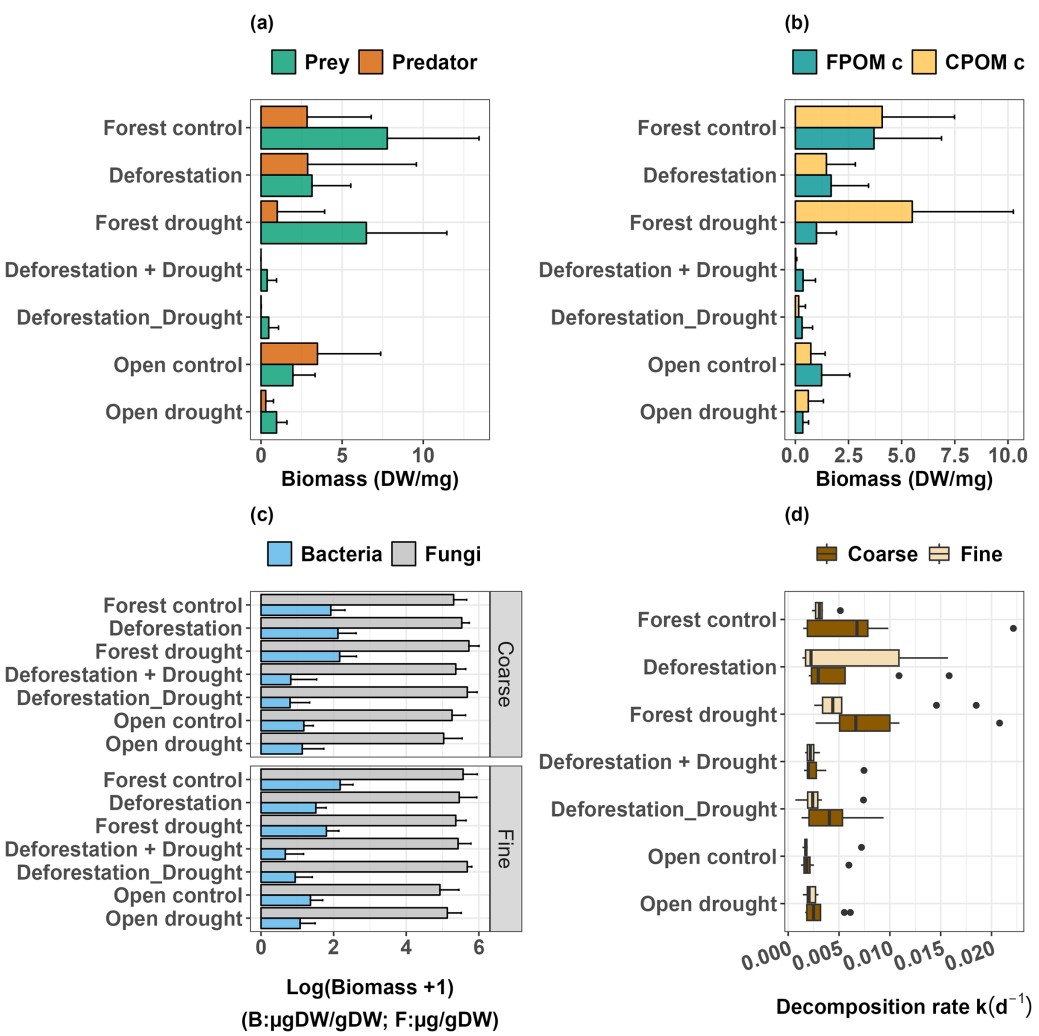

**Figure 3 Response of freshwater communities and ecosystem function to the simulated extreme drought, the transplantation and the sequential and simultaneous combination of both stressors.** (A) Prey and predator biomass, (B) FPOM and CPOM consumer biomass, (C) biomass of micro-organisms (fungi and bacteria) attached to leaf litter, (D) leaf litter decomposition.

## Biomass of bacteria and fungi at the surface of the leaf litter

The biomass of attached bacteria in control bromeliads was higher in the forest than in the open area, whatever the mesh size of enclosures (Coarse, Mann-Whitney U: $W = 87$, $p < 0.0100$; Fine, Mann-Whitney U: $W = 52$, $p < 0.0500$). The biomass of fungi was not significantly different across areas in the coarse mesh enclosures, but was higher in the forest in fine mesh enclosures, compared to the open area ($p < 0.0500$, Table 2, Fig. 3C, Table S3 and Fig. S3). Deforestation and drought alone did not induce significant changes in the biomass of bacteria (Table 2, $p > 0.0500$ for all tests; Fig. 3C; see also Table S3 for control *vs* treatments in the open area). The interactive effects of deforestation and drought significantly reduced the biomass of attached bacteria for both mesh sizes (Fig. 3C, Table 2, sequential deforestation and drought, Coarse: $p < 0.0010$, Fine: $p < 0.0001$; simultaneous

**Table 1** *Post-hoc* dunnett tests with Bonferroni correction, comparing the biomass of the various invertebrate feeding groups in control *vs* treatment bromeliads.

| Univariate response | Control *vs* treatment | Estimate | SE | DF | Lower.CL | Upper.CL | T ratio | *P* value |
|---|---|---|---|---|---|---|---|---|
| Predators | FC – Deforestation | −5.3 | 3.69 | 45 | −14.9 | 4.29 | −1.437 | 0.6301 |
| | FC – Forest drought | −16.5 | 3.69 | 45 | −26.1 | −6.91 | −4.475 | **0.0002** |
| | FC – D + Dr | −27.6 | 3.69 | 45 | −37.2 | −18.01 | −7.485 | **<0.0001** |
| | FC – D_Dr | −26.1 | 3.69 | 45 | −35.7 | −16.51 | −7.079 | **<0.0001** |
| Prey | FC – Deforestation | −10.35 | 3.85 | 45 | −20.4 | −0.336 | −2.689 | **0.0400** |
| | FC – Forest drought | −2.15 | 3.85 | 45 | −12.2 | 7.864 | −0.559 | 1.0000 |
| | FC – D + Dr | −28.05 | 3.85 | 45 | −38.1 | −18.036 | −7.289 | **<0.0001** |
| | FC – D_Dr | −26.70 | 3.85 | 45 | −36.7 | −16.686 | −6.938 | **<0.0001** |
| FPOM consumers | FC – Deforestation | −9.9 | 5.08 | 45 | −23.1 | 3.31 | −1.950 | 0.2298 |
| | FC – Forest drought | −12.1 | 5.08 | 45 | −25.3 | 1.16 | −2.373 | 0.0879 |
| | FC – D + Dr | −24.9 | 5.08 | 45 | −38.1 | −11.69 | −4.904 | **0.0001** |
| | FC – D_Dr | −25.1 | 5.08 | 45 | −38.4 | −11.94 | −4.953 | **<0.0001** |
| CPOM consumers | FC – Deforestation | −6.65 | 3.67 | 45 | −16.20 | 2.9 | −1.812 | 0.3065 |
| | FC – Forest drought | 4.00 | 3.67 | 45 | −5.55 | 13.5 | 1.090 | 1.0000 |
| | FC – D + Dr | −26.05 | 3.67 | 45 | −35.60 | −16.5 | −7.099 | **<0.0001** |
| | FC – D_Dr | −27.80 | 3.67 | 45 | −31.35 | −12.3 | −5.941 | **<0.0001** |

Note:
FC, forest control; D + Dr, deforestation and drought sequentially; D_Dr, deforestation and drought simultaneously; *p*-value in bold denote significant effects ($p < 0.05$). Estimate, contrast of means; SE, standard error; DF, degree of freedom; Lower and Upper CL, confidence limits for the mean.

**Table 2** *Post-hoc* dunnett tests with Bonferroni correction, comparing microorganism biomass in control vs treatment bromeliads.

| Univariate response | | Control *vs* treatment | Estimate | SE | DF | Lower. CL | Upper. CL | T ratio | *P* value |
|---|---|---|---|---|---|---|---|---|---|
| Attached bacteria | Coarse | FC – Deforestation | 1.26 | 4.51 | 44 | −10.50 | 13.01 | 0.278 | 1.0000 |
| | | FC – Forest drought | 3.20 | 4.39 | 44 | −8.24 | 14.64 | 0.729 | 1.0000 |
| | | FC – D + Dr | −20.80 | 4.39 | 44 | −32.24 | −9.36 | −4.736 | **0.0001** |
| | | FC – D_Dr | −19.30 | 4.39 | 44 | −30.74 | −7.86 | −4.395 | **0.0003** |
| | Fine | FC – Deforestation | −9.83 | 4.42 | 34 | −21.5 | 1.83 | −2.224 | 0.1316 |
| | | FC – Forest drought | −5.28 | 4.04 | 34 | −15.9 | 2.37 | −1.308 | 0.7991 |
| | | FC – D + Dr | −23.70 | 3.95 | 34 | −34.1 | −13.27 | −5.993 | **<0.0001** |
| | | FC – D_Dr | −18.00 | 4.14 | 34 | −28.9 | −7.09 | −4.352 | **0.0005** |
| Attached fungi | Coarse | FC – Deforestation | 8.31 | 6.03 | 44 | −7.394 | 24.0 | 1.378 | 0.7003 |
| | | FC – Forest drought | 16.10 | 5.87 | 44 | 0.813 | 31.4 | 2.743 | **0.0351** |
| | | FC – D + Dr | 1.50 | 5.87 | 44 | −13.787 | 16.8 | 0.256 | 1.0000 |
| | | FC – D_Dr | 15.10 | 5.87 | 44 | −0.187 | 30.4 | 2.573 | 0.0542 |

Note:
FC, forest control; D + Dr, deforestation and drought sequentially; D_Dr, deforestation and drought simultaneously. *p*-value in bold denote significant effects ($p < 0.05$). Estimate, contrast of means; SE, standard error; DF, degree of freedom; Lower and Upper CL, confidence limits for the mean.

deforestation and drought, Coarse: $p < 0.0010$, Fine: $p < 0.0010$). In fine mesh enclosures, the biomass of fungi did not vary significantly across treatments (aligned rank Transform ANOVA: Df = 4, F value = 1.674, $p > 0.0500$). In coarse mesh enclosures however, fungal biomass was significantly boosted by drought and marginally increased by the simultaneous impact of deforestation and drought (Drought: $p < 0.0500$; simultaneous

**Table 3 Post-hoc dunnett tests with Bonferroni correction, comparing decomposition rate in control vs treatment bromeliads.**

| Univariate response | | Control vs treatment | Estimate | SE | DF | Lower. CL | Upper. CL | T ratio | P value |
|---|---|---|---|---|---|---|---|---|---|
| Decomposition rate | Coarse | FC – Deforestation | −1.02 | 6.00 | 44 | −16.66 | 14.61 | −0.170 | 1.0000 |
| | | FC – Forest drought | 8.70 | 5.84 | 44 | −6.52 | 23.92 | 1.489 | 0.5745 |
| | | FC – D + Dr | −12.20 | 5.84 | 44 | −27.42 | 3.02 | −2.088 | 0.1705 |
| | | FC – D_Dr | −4.40 | 5.84 | 44 | −19.62 | 10.82 | −0.753 | 1.0000 |
| | Fine | FC – Deforestation | −6.50 | 5.69 | 34 | −21.50 | 8.50 | −1.143 | 1.0000 |
| | | FC – Forest drought | 6.00 | 5.19 | 34 | −7.69 | 19.69 | 1.156 | 1.0000 |
| | | FC – D + Dr | −11.20 | 5.09 | 34 | −24.61 | 2.21 | −2.202 | 0.1380 |
| | | FC – D_Dr | −7.38 | 5.32 | 34 | −21.40 | 6.65 | −1.387 | 0.6982 |

**Note:**
FC, forest control; D + Dr, deforestation and drought sequentially; D_Dr, deforestation and drought simultaneously. Estimate, contrast of means; SE, standard error; DF, degree of freedom; Lower and Upper CL, confidence limits for the mean.

deforestation and drought $p > 0.0500$, Table 2, Figs. 3C and S3). Last, neither the biomass of bacteria nor fungi varied with the emulated drought in the open area (see Table S3).

**Leaf litter decomposition**

The comparison of control bromeliads (open vs forest) showed that decomposition in the open area was significantly lower than in the forested area (Coarse, Mann-Whitney U: W = 77, $p < 0.0500$; Fine, Mann-Whitney U: W = 54, $p < 0.0100$; Fig. 3D). There was no significant effect of mesh size on decomposition rates in control groups, meaning that decomposition was mostly driven by microbial activity. Overall, there was no significant effect of deforestation, drought and deforestation x drought on the decomposition rates, whatever the mesh size of enclosures (Table 3). Finally, the comparison of treatment and control bromeliads in the open area did not show a significant effect of drought on the decomposition rates (Coarse, Mann-Whitney U: W = 26, $p > 0.0500$; Fine, Mann-Whitney U: W = 26, $p > 0.0500$).

## DISCUSSION

We found that deforestation and drought alone had a moderate impact on the aquatic ecosystem of tank bromeliads, but that their combination had deleterious, synergistic effects on multitrophic communities. In line with our first and second hypotheses, deforestation negatively affected detritivore biomass (but did not eliminate species, see Fig. S2) and drought significantly reduced predator biomass in both forest and open areas, which could induce a change in the overall structure of multitrophic communities (see also *Romero et al., 2020*). Though we expected litter decomposition to be affected by changes in invertebrate biomass, we did not detect food-web mediated effects of drought or deforestation alone on decomposition. These results suggest that individual species and the aquatic ecosystem of tank bromeliads are relatively resistant to each of these disturbance types. Our third hypothesis of a stronger effect of combined disturbances was verified, as the combination of deforestation and drought synergistically reduced the biomass of all invertebrate functional groups and bacteria, whatever the sequence of the disturbance

types. Among the various trophic levels, predators were virtually eliminated while detritivore biomass was strongly depleted (Table 1, Figs. 3A and 3B). Nevertheless, contrary to our assumptions, decomposition was not sensitive to any of our treatments involving deforestation and/or drought (Table 3 and Fig. 3D). We note that fungi, key players of the decomposition process (*Pascoal & Cássio, 2004*), were either unaffected or boosted by our treatments (Table 2 and Fig. 3C). As decomposition was essentially microbial in treatment and control groups, our results further suggest that a core group of highly-resistant microorganisms plus a few invertebrate detritivores (among which Tipulidae are shredders), ensure the stability of key ecosystem functions across environments (*Dézerald et al., 2013*), and in the face of local to global environmental change. Moreover, our experiment demonstrates that even an intrinsically-resistant Neotropical ecosystem, that has the capacity to maintain its structure and functioning when exposed to a single disturbance type, can experience dramatic changes in food web structure when exposed to a combination of local- and global-scale disturbances. Assuming that taking global action is more challenging than taking local-regional actions, policy-makers should be encouraged to implement environmental action plans that will halt habitat destruction, to dampen any detrimental interactive effect with the impacts of global climate change.

We see at least two non-mutually exclusive explanations for the weak effects of deforestation or drought on aquatic ecosystems: (i) the ability of physical habitats (here the bromeliad rosette) to buffer changes in hydrology and resource inputs (*Fernandez Barrancos, Reid & Aronson, 2017*), and (ii) the ability of organisms to behaviourally and/or physiologically cope with these changes. First, tank bromeliads have evolved to capture and retain water and detritus so the system itself is relatively resistant to dry periods, and able to collect various types of palatable detritus in any habitat type (litter, dust, dead insects, faeces, *etc.*). The multiple leaves divide the overall water volume, thus reducing evaporation rates during a drought, or despite changes in canopy cover that alter ambient humidity and throughfall. For instance, it takes up to 7 weeks to completely dry out a small bromeliad in the forest in the absence of rainfall (*Dézerald et al., 2015*). Here, only three out of 20 bromeliads subjected to drought completely dried out, and all bromeliads subjected to deforestation kept a water volume superior to 100 mL, the threshold for the presence of large odonate larvae (predators) in tank bromeliads (*Srivastava et al., 2020*). This physical buffering effect that benefits to aquatic micro- to macro-organisms can be extended to other lentic freshwater ecosystems such as ponds and pools, where habitat complexity (*e.g.*, surface:volume ratio, porosity, drainage basin area) dampen the effects of environmental change (*Kebede et al., 2006*). Second, species tend to be generalists and/or have evolved tolerance and resistance traits that allow ecological adaptation to changing conditions (*Dézerald et al., 2015*; *Strachan, Chester & Robson, 2015*). We know for instance that bromeliads host a core of generalist species (including invertebrates and microorganisms) that is relatively constant across Neotropical environments (open savannah, primary and secondary forest, plantations) and across large regional scales, so that plasticity in basal resource use is widespread and shifts in food web structure are mostly related to the addition or loss of top-predators (*Dézerald et al., 2013*; *Romero et al.,*

*2020*). A more recent study further showed that the median lethal time (LT50) of bromeliad invertebrate populations experimentally subjected to desiccation varies from 4 to 19 days depending on the species. The literature and our results therefore support the idea that the plant-held waters of Neotropical forests host multitrophic communities that are resistant to drought and deforestation, provided that these changes are not combined.

The interactive effect of deforestation and drought was much higher than the sum of individual effects, for entire invertebrate populations or even functional groups (*e.g.*, predators) were wiped out, and the biomass of bacteria decreased significantly when it was not affected by each disturbance in isolation (Tables 1 and 2; Fig. 3). This result means that a negative synergism (*sensu Piggott, Townsend & Matthaei, 2015*) of two disturbances that each have moderate effects drove mortality beyond the physical and biological buffering capacities of the system. Synergistic interactions could be generated by different types of effects: (i) interaction chain effects (*Buma, 2015*), *e.g.*, if deforestation enhances drought severity by increasing evaporation rates (ii) interaction modification effects (*Trzcinski et al., 2016*), *e.g.*, if the *per capita* effects of deforestation on aquatic organisms is enhanced by drought, and (iii) interaction exposure effects, *e.g.*, if preceding deforestation altered the taxonomic and/or functional structure of communities further exposed to drought (*Shinoda & Akasaka, 2020*). As the outcomes of simultaneous and sequential disturbances were not different, we can reasonably exclude exposure effects. Because the outcome of the interaction was to dry out bromeliads (something that did not happen with drought or deforestation alone), we further conclude that most of the observed impacts on multitrophic communities can be attributed to interaction chain effects.

Whilst bacterial biomass declined under the interactive effects of deforestation and drought, fungi were resistant to even the harshest experimental conditions emulated. In presence of invertebrates, fungal biomass was even boosted by severe disturbances (Table 2, Figs. 3 and S3). Similar observations in different ecosystems type and at different latitudes (*Li et al., 2023*; *Yuste et al., 2011*), lead to the general conclusion that fungi are much more tolerant to drought than bacteria because they evolved desiccation-tolerance traits such as thick cell walls and osmolytes (*Treseder et al., 2010*). Interestingly, fungi can replace bacteria in the decomposition process under high stress conditions (*Yuste et al., 2011*). Assuming that decomposition in tank bromeliads is mostly driven by microbial activity (*Leroy et al., 2017*; this study), and that fungi can remain active under extremely dry conditions (*Allison et al., 2013*), it is therefore not surprising that decomposition did not decline in manipulated bromeliads (see also *Schaeffer et al., 2017*; *Schimel, Balser & Wallenstein, 2007*), including the driest conditions created by drought x deforestation. Together with the literature, our results support the ideas that (i) fungal hyphae play a crucial role for the persistence of detrital decomposition, a key ecological process in Neotropical ecosystems and in other ecosystems worldwide, in the face of environmental change (*Banerjee et al., 2016*; *Philippot, Griffiths & Langenheder, 2021*), and (ii) a drier world resulting from the interaction of habitat destruction and global climate change might favour microbial communities dominated by fungi (*Treseder et al., 2018*). We finally note that a few invertebrate species were able to maintain populations under interactive effects, namely *Aulophorus superterrenus* (Annelid, Oligochaeta), *Elpidium bromeliarum*

(Crustacea, Ostracoda), an unidentified Chironomini (Insecta, Chironomidae), and *Trentepohlia* sp. (Insecta, Tipulidae). These species are among the commonest, both in terms of occurrence and number of individuals, in tank bromeliads of French Guiana whatever the habitat type (*Dézerald et al., 2017*; *Jabiol et al., 2009*). This suggests that only ubiquist species will resist the interactive effects of deforestation and drought while both frequent and rare species will be eliminated, probably to the detriment of unique functional trait combinations (*Céréghino et al., 2022*).

## CONCLUSIONS

To the best of our knowledge, this study is the first attempt to understand the combined effect of drought and deforestation at the level of an entire freshwater ecosystem in the Neotropics, a region of the World that both hosts species-rich ecosystems and is particularly subject to these disturbances. We acknowledge that our model system, the bromeliad ecosystem, is quite resistant in the face of deforestation or drought (see also *Bonhomme et al., 2021*). In other freshwater ecosystems such as streams for instance, either drought or deforestation can lead to much stronger changes in decomposition rates following to a decline in shredders diversity and density (*e.g.*, *Lake, 2003*; *Lima et al., 2022*; *Silva-Araújo et al., 2020*; *Tanaka et al., 2015*). However, if these systems are more sensitive than bromeliads to either deforestation or drought, we can then conclude from our results that, even if drought or deforestation have moderate or even short-term impacts, a combination of the two disturbances will generally have deleterious effects on freshwater biodiversity and ecosystem functions (*Taniwaki et al., 2017*). Beyond the ecological implications, the results and conclusions reported in this study provide important insights to decision makers. Not only global climate change and habitat destruction operate at different spatial and temporal scales, but political agreements of global extent are much more difficult to obtain than regional-national actions (*Galaz, 2022*). Therefore, whilst it would take decades or centuries to counteract the negative effects of climate change, action could theoretically be taken to halt habitat destruction with the aim to mitigate the drastic effects local x global environmental change on freshwater ecosystems.

## ACKNOWLEDGEMENTS

The authors would like to thank Heidy Schimann, Eliane Louisanna, Valérie Troispoux, Justine Renaud, and Mathieu Gallant Canguilhem for their help in the field. We also thank Hermine Billard and Jonathan Colombet, Platerforme CYSTEM-UCA PARTNER (Clermont-Ferrand, FRANCE), for their technical support and expertise. Two reviewers provided helpful comments on an earlier version of the manuscript.

### Funding

This work received financial support from an Investissement d'Avenir grant managed by the Agence Nationale de la Recherche (CEBA, ref. ANR-10-LABX-25-01), from a CNRS-EC2CO Grant (project AQUATROP) and from French Agence Nationale de la Recherche

(ANR) through the Resilience project (Grant ANR-18-CE02-0015). The funders had no role in study design, data collection and analysis, decision to publish, or preparation of the manuscript.

### Grant Disclosures
The following grant information was disclosed by the authors:
Agence Nationale de la Recherche: CEBA, ref. ANR-10-LABX-25-01.
CNRS-EC2CO: AQUATROP.
French Agence Nationale de la Recherche (ANR): ANR-18-CE02-0015.

### Competing Interests
The authors declare that they have no competing interests.

### Author Contributions

- Marie Séguigne performed the experiments, analyzed the data, prepared figures and/or tables, authored or reviewed drafts of the article, and approved the final draft.
- Céline Leroy conceived and designed the experiments, performed the experiments, authored or reviewed drafts of the article, and approved the final draft.
- Jean-François Carrias conceived and designed the experiments, performed the experiments, authored or reviewed drafts of the article, and approved the final draft.
- Bruno Corbara conceived and designed the experiments, performed the experiments, authored or reviewed drafts of the article, and approved the final draft.
- Tristan Lafont Rapnouil performed the experiments, authored or reviewed drafts of the article, and approved the final draft.
- Régis Céréghino conceived and designed the experiments, performed the experiments, authored or reviewed drafts of the article, and approved the final draft.

### Field Study Permissions
The following information was supplied relating to field study approvals (*i.e.*, approving body and any reference numbers):
French Ministry of Ecological Transition and Solidarity.

### Data Availability
The code and data are available in the Supplemental Files.

### Supplemental Information
Supplemental information for this article can be found online at http://dx.doi.org/10.7717/peerj.17346#supplemental-information.

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
