# Peer review of "Interactive effects of drought and deforestation on multitrophic communities and aquatic ecosystem functions in the Neotropics—a test using tank bromeliads"

_PeerJ, doi:10.7717/peerj.17346_

## Round 0.1 · original submission · Minor Revisions

I have now received two reviews of the manuscript. I found the manuscript very interesting, which is something also highlighted by both reviewers. However, reviewers and I found some aspects of the manuscript that should be corrected before it can be accepted for publication.

Reviewer #1 asks for some specific clarifications of statistical procedures and more discussion regarding the three hypotheses and predictions presented in the introduction.

Reviewer #2 asked for some clarification on the experimental design and also expressed concern about the statistical analyses, something I agree with.

After reading and analyzing the ms, I have some concerns about two aspects of the statistical section of the study. On the one hand, you highlight the interaction among factors (deforestation x drought), however, you also stated that data didn’t fulfill ANOVA’s requirements so you applied mean comparisons (U test). You need to clarify how you analyzed the interactions and how you managed the errors when performing the multiple comparison tests (see also comments of reviewer #2). If these corrections were included in the R package, you need to clarify this in the text, as readers may not be familiar with the specific package. On the other hand, in the discussion several times you referred to marginal differences (also highlighted by reviewer #1), but you never mentioned them in the results. You need to define in the Method section what was considered marginal and clarify this in the results section.

·

Basic reporting

No comment

Experimental design

No comment

Validity of the findings

The language of the discussion and conclusion in some cases does not stem from the results presented. This does not seem to be due to mistakes in methodology or statistical reasoning, but some changes may need to be made to the text to improve clarity.

Additional comments

Methods
178 – What do you mean by “similar”? It might be good to list a range.
181 – “First, twenty bromeliads of the forested area were transplanted” After how long of a time spent in the forested area?
187 – Why say half of the transplanted bromeliads when that’s 10 bromeliads. It would be more clear to say something like “ten bromeliads in the forest and in the open area as well as 10 transplanted bromeliads…”
Figure 1 should be sited in the text at some point.
205 – What are you referring to when you say plant content? This is the water in the tank? You never refer to it as that again in the paper?
207 – Was there anything else in the tank besides water and the detritus you added (e.g., invertebrates)? Would be good to very quickly clarify what “whole content” includes.
233 – Do you have any source for these delineations?
241- Can you briefly list the “usual data transformations”?
250 – Should (T1 to T5) go after “against forest control” in the text?
266 – “models did not met assumptions of the Anova method used” based on what method?

Results
273- Wouldn’t it be clearer and more accurate to say, “Control bromeliads held more water compared to drought treated bromeliads”? I am not sure why you say “different”.
278 – “Dried out 99% of all transplanted bromeliads” This can’t be right. There were not 100 bromeliads. Where does that number come from? Are you trying to say all but one of the bromeliads in that treatment were dried out?
281 – Extra parenthesis/missing parenthesis somewhere “p < 0.001)”
286 – Table S1 is out of order as it comes after table S2 in the text, other tables appear to be out of order as well.
291– What test is the p value derived from here? You go back and froth between it and the Mann Witney U statistics
293 – “(Forest: Table 1; Open area” I am not familiar with this formatting, but neither forest nor open area appear in the text of table 1. Is this two tests? Perhaps this is a typo, or you need to change the text here to match the table?
298- What do you mean by “maintain populations in bromeliads”? They were the only species found in these plants? Had a population above a certain threshold? If so, you should more directly state that.
302- Please describe somewhere in the methods that you further categorized detritivores by ingested particle size and how that was determined.

Discussion
You introduce 3 hypotheses in the introduction, but never reference them directly in the discussion or conclusion. It would be helpful for the reader to restate these hypotheses and state if your results supported them.
Several claims are made in the discussion, without citing the corresponding result. It is helpful for the reader to be able to see which result (table or figure) you are referencing when making a claim about your results (e.g., line 350-352).
345 – What you found was drought reduced the biomass of predators. This could lead to changes in the overall structure of multitrophic communities, but that is not what you measured or tested for explicitly. It would be good to change the sentence to reflect that.
362- Where is the evidence that the system “collapsed”? Your previous point was that neither treatment effected decomposition. I do not see any evidence of collapse in any of your data.
367- I am not clear on which “marginal effects” you are referring to here? You last paragraph described both significant and non-significant effects as well as a description of ecosystem “collapse” which is not what I would describe as “marginal”. If you are trying to explain the lack of response, make sure you are clear about which response you are referring to.

Figures

Figure 1 – If there was a transplantation at day zero then what were the conditions for the plants leading up to day zero?

Reviewer 2 ·

Basic reporting

This is an interesting paper with a nice overall approach. The authors investigated isolated and interactive effects of drought and deforestation on the community structure of macro- and microorganisms and an important ecosystem function (ecosystem decomposition). They reported that drought or deforestation alone had a moderate impact on biomass at the various trophic levels. However, their interaction synergistically reduced the biomass of invertebrate functional groups and bacteria. Fungal biomass was either unaffected or boosted by the treatments. Finally, the decomposition was not affected by the treatments.
Overall, this is a well-conducted manuscript, but I have some concerns.

Experimental design

My most important concerns are related to the experimental design, which should be revised carefully. Some deficiencies in the statistical analysis should also be improved or reevaluated to sustain the study's results better.
I) I'm confused and worried about how the experiment was designed to investigate the effect of drought and deforestation. The authors mention that initially, ten bromeliads were transplanted from a closed area to an open area. After 105 days, the experimental effect of drought began. Another set of bromeliads was transplanted from the forested area to the open area, and the effects of drought were promptly applied to these bromeliads.
Some questions:
a) Were the bromeliads transplanted intact, with the fauna and flora of the original area in both cases?
b) Transplanting bromeliads from one area to another can cause physiological stress on the plants. Thus, a plant that remained in place for 105 days before applying the experimental effect can recover from this stress, while a newly transplanted plant can still be under this transplant stress. How can the authors assume that possible differences between these two treatments and the others (deforestation followed by drought and deforestation and drought simultaneously) are due to the physiological stress caused by the removal and planting of the plants and not by the treatment per se? It would be necessary for the authors to explain or clarify these points. Still considering physiological stress, some bromeliads were transplanted, and others were not. Therefore, some bromeliads may have been subjected to the stress of removal and planting, while others may not. Hence, these physiological stresses were not standardized among plants in all treatments, which could cause a confusing effect between treatments and the potential stress caused by handling bromeliads. These are essential points that can compromise the robustness of the experimental effects.
c) The fauna and flora of bromeliads have a rapid life cycle. Thus, after 170 days of experimentation, much of the bromeliad fauna and flora from the new location that was natural to the area of origin (e.g., bromeliads from forested areas transplanted to open areas) may have been replaced by biota from the new area (open area). Thus, the experimental effect of deforestation may be weakened by this scenario. How did the authors consider this?

Validity of the findings

II) Regarding the statistical analyses, the authors conducted multiple tests and comparisons between treatments, which may increase the chances of a Type I error occurring.
The fundamental problem is that when multiple tests are undertaken, each at the same significance level (e.g., P<0.05), the probability of achieving at least one significant result is greater than that significance level (Rice 1989). Therefore, there is an increased probability of rejecting a null hypothesis when it would be inappropriate to do so. Common solutions to these problems are the use of Bonferroni correction to the results (Rice 1989) or reporting effect size and/or confidence intervals for effect size (or other alternatives) (Nakagawa 2004).
Rice WR, 1989. Analyzing tables of statistical tests. Evolution43:223-225.
Shinichi Nakagawa, A farewell to Bonferroni: the problems of low statistical power and publication bias, Behavioral Ecology, Volume 15, Issue 6, November 2004, Pages 1044–1045, https://doi.org/10.1093/beheco/arh107

Additional comments

Specific comments
L121-122 – The loss of canopy cover can increase luminosity and, consequently, primary productivity and algal blooms, thus also affecting the green food web pathway. Perhaps this can also be considered in predictions.
L157-158 - Was leaf detritus collected at the study site? Does the plant species in question commonly occur in the study area? Please include this information.
L178-179 - What is the average and variation of these vegetative traits? Please include this information.
L184—This drought event appears much larger than the average reported above by the authors (26 ± 5.3 days). Please double-check this information.
Results:
- Please standardize the number of decimal places used to describe P values.
- Tables 1 and 2—Please provide more detail on the statistical parameters reported in Tables 1 and 2 and the significance level used in the tests.
L298 - Did you mean Figure 3a is from the main text or supplemental material?

---

## Round 0.2 · accepted · Accept

After carefully reading the answers to reviewers, my editorial comment, and the new version of the manuscript I found that all the comments were satisfactorily addressed. Thus, the manuscript is accepted for publication as it is in this new revised version.